# A Bioactive Gelatin-Methacrylate Incorporating Magnesium Phosphate Cement for Bone Regeneration

**DOI:** 10.3390/biomedicines12010228

**Published:** 2024-01-19

**Authors:** Xiping Zhang, Changtian Gong, Xingyu Wang, Zhun Wei, Weichun Guo

**Affiliations:** Department of Orthopedics, Renmin Hospital of Wuhan University, Wuhan 430060, China; 2016302180026@whu.edu.cn (X.Z.); gongct@whu.edu.cn (C.G.); 2017302180019@whu.edu.cn (X.W.); weizhun1998@whu.edu.cn (Z.W.)

**Keywords:** GelMA hydrogel, bone regeneration, MPC magnesium

## Abstract

Maintaining proper mechanical strength and tissue volume is important for bone growth at the site of a bone defect. In this study, potassium magnesium phosphate hexahydrate (KMgPO_4_·6H_2_O, MPC) was applied to gelma-methacrylate hydrogel (GelMA) to prepare GelMA/MPC composites (GMPCs). Among these, 5 GMPC showed the best performance in vivo and in vitro. These combinations significantly enhanced the mechanical strength of GelMA and regulated the degradation and absorption rate of MPC. Considerably better mechanical properties were noted in 5 GMPC compared with other concentrations. Better bioactivity and osteogenic ability were also found in 5 GMPC. Magnesium ions (Mg^2+^) are bioactive and proven to promote bone tissue regeneration, in which the enhancement efficiency is closely related to Mg^2+^ concentrations. These findings indicated that GMPCs that can release Mg^2+^ are effective in the treatment of bone defects and hold promise for future in vivo applications.

## 1. Introduction

Normal conditions allow bone to regenerate rapidly. Throughout adulthood, the microenvironment surrounding bone—including inflammatory cells, endothelial cells, and Schwann cells—continues to facilitate repair processes that restore damaged bone tissue to a homeostatic functional state [1,2,3]. However, this self-healing method is only suitable for repairing relatively small bone defects [4,5]. For large bone defects caused by infection, trauma, and the resection of bone tumors, the normal physiological structure of bone cannot be restored without medical intervention [6,7,8]. The best clinical methods to solve bone defects mainly include autologous bone graft, allograft bone graft, and xenograft bone graft, among which autologous bone graft is considered to be the gold standard of bone graft. However, autologous bone graft has great limitations in bone extraction, and the donor site is prone to infection and massive blood loss [2,9,10].

Gelatin is an inducible protein derived from collagen, the natural protein that constitutes the skin, tendons, and cartilage of animals. It possesses remarkable attributes such as exceptional biodegradability, biocompatibility, and non-toxicity [11,12]. Gelatin/methacrylic anhydride (GelMA), formed by introducing methacrylate groups to amine groups in gelatin, can be transformed into photo-crosslinked hydrogels [13,14].

Many studies have proved that GelMA hydrogels have good long-term biocompatibility [15]. In addition, GelMA hydrogels have great advantages in bioengineering fields such as cell culturing, tissue regeneration, and drug delivery due to their similarity to the extracellular matrix [16]. At the same time, the mechanical strength of hydrogels is lower than that of other materials, and other substances need to be added to improve its mechanical strength if it is applied to tissue regeneration [17,18].

Calcium phosphate cement (CPC) has been used in bone reconstruction for many decades because it mimics bone’s inorganic nature [19,20]. Although CPC has good biocompatibility and bone conductivity, it also has certain limitations, among which are coagulation times that are prolonged, mechanical properties that are relatively poor, and degradation rates that are slow [21,22,23].

A magnesium phosphate cement (MPC) with relatively high mechanical properties and a decrease in degradation rate is evaluated in this context [24,25]. About 67% of magnesium is found in bone tissue, making it the fourth most abundant metal element in the body [26]. More than 600 enzymatic reactions can be affected by magnesium ions (Mg^2+^), including energy metabolism and protein synthesis [27]. Magnesium ions (Mg^2+^) can affect more than 600 enzymatic reactions, including those related to energy metabolism and protein synthesis. In bone tissue, Mg^2+^ can be incorporated into the bone mineral lattice to regulate the biomechanics of natural bone [28]. Compared with CPC, MPC has faster solidification speed, higher strength, and promotes bone formation [29,30]. Traditional MPC ammonium is produced through the hydration process of ammonium dihydrogen phosphate (NH_4_H_2_PO_4_) and dead-burned magnesium oxide (MgO). It should be noted, however, that ammonia can be cytotoxic during the reaction, making ammonia MPC unsuitable for clinical applications [9]. As a result of this limitation, potassium dihydrogen phosphate (KH_2_PO_4_) has been used to replace NH_4_H_2_PO_4_ in order to avoid the release of ammonia. Based on the hydration reaction, MgO is the alkaline substrate and KH_2_PO_4_ is the acidic substrate in the MPC reaction. By introducing deionized water into the powder phase, KH_2_PO_4_ dissolves in the liquid, releasing K^+^, H^+^, and PO_4_^3−^. By reacting with H_2_O and H^+^, MgO releases Mg^2+^ that reacts with dissolved ions to produce MgKPO_4_·6H_2_O. As a final step, MgKPO_4_·6H_2_O forms a coordinated network that consolidates around the unreacted MgO to form a hardened cement [9,31]. Although potassium salt MPC has many advantages, its setting time is short and most of it is prepared in columnar form, which brings difficulties to practical application [32]. Therefore, by grinding it into powdery particles combined with bioactive substances, the specific surface area of MPC can be further increased, thus increasing the possibility of clinical use [2].

To sum up, we prepared a new GELMA-MPC composite (GMPC) by mixing MPC and GelMA hydrogels in different proportions, ensuring that the material retains its shape while promoting bone healing. In this study, we explored and analyzed the physical and chemical properties of the prepared GelMA and GMPC and demonstrated their biocompatibility and bone regeneration through in vitro and in vivo experiments with rat bone marrow mesenchymal stem cells (MSC), human umbilical vein vascular endothelial cells (HUVEC), and rat models (Figure 1).

## 2. Materials and Methods

### 2.1. Preparation of GelMA/MPC Composites

GelMA was synthesized as previously reported [18]. In brief, 20 mL of PBS was put in a brown bottle containing 0.05 g Lithium Phenyl (2,4,6-trimethylbenzoyl) phosphinate (LAP, Engineering For Life), followed by dissolution in a water bath at 40–50 °C for 15 min. Then, gelatin (GelMA-60 (degree of amino substitution: 60 ± 5%, molecular weight: 100–200 kDa, turbidity: <20%)) was dissolved in LAP solution with a concentration of 5% (*w*/*v*), followed by heating at 60–70 °C for 25 min in the dark. Finally, a basic GelMA solution (5%, *w*/*v*) was successfully prepared.

MgO and KH_2_PO_4_ were purchased from Sinopharm Chemical Reagent Co., Ltd. (Shanghai, China). MPC was prepared using powder and liquid phases. With a purity of 98.5% of dead-burned MgO style and a purity of 99.5% KH_2_PO_4_, the mole ratio was 1.5:1 in the powder preparation phase. The initial MgO was calcined at 1600 °C for 3 h, and the dead-burned MgO and KH_2_PO_4_ were ground in a planetary ball mill and sieved through a 200-mesh sieve for 2 h. The selected MgO and KH_2_PO_4_ powders were mixed at a molar ratio of 1.5:1 and deionized water was added at a ratio of 2 g mL^−1^ [21]. After self-setting and demolding, MPC powders were obtained through grinding and being passed through another 200-mesh sieve for 2 h.

MPC powders were mixed with the basic GelMA (5%, *w*/*v*) on the basis of weight-to-volume ratio (g ml mL^−1^, 2.5%, 5%, and 7.5%). Then, UV curing was performed using mercury arc lamp (405 nm, 20 s) to prepare cross-linked GelMA hydrogel, namely, 2.5 GMPC, 5 GMPC and 7.5 GMPC, respectively. Then, basic GelMA was set as control.

### 2.2. Microstructure Analysis

For further experiments, the diameter of the crosslinked composite was cut to 6 mm. The obtained GelMA, 2.5 GMPC, 5 GMPC and 7.5 GMPC samples were freeze-dried in the freeze-drying machine for 36 h. The surface microstructures of GelMA and GMPCs were examined via high-resolution scanning electron microscopy (SEM) DSM 940 (Zeiss, Jena, Germany) after freeze-drying. Before examination, all specimens were stuck on special holders via conductive stickers and then sputtered with a thin (4 nm) gold layer for electron reflection.

### 2.3. X-ray Diffraction

The XRD patterns were obtained using an X-ray diffractometer (Smart Lab SE, Tokyo, Japan). Film samples with dimensions of 5 mm × 5 mm were cut and fixed in a circular clamp of the instrument. The analysis was carried out directly and the conditions were as follows: (1) voltage and current: 40 kV and 40 mA, respectively; (2) scan range from 10° to 70°; (3) step: 0.1 d°; and (4) speed 1 d°/min. This procedure involved a secondary monochromator of graphite beams. The samples were stored at 25 °C and 50% relative humidity (RH) and analyzed in triplicate.

### 2.4. Infrared Analysis

The chemical compositions of GelMA and GMPCs were assessed via Fourier transform infrared spectroscopy with attenuated total reflectance (FTIR-ATR) using an FTIR 5700 (Thermo Electron Corporation, Fitchburg, WI, USA) spectrometer equipped with a diamond crystal at a nominal incidence angle of 45° and ZnSe lens. Spectra were recorded in the range of 600–4000 cm^−1^ at 32 scans with a resolution of 4 cm^−1^.

### 2.5. In Vitro Mechanical Analysis of GMPCs

We used a rotational rheometer under 25 °C for the circular mold preparation of various GelMA hydrogels and performed GMPC sample measurement to research frequency-dependent viscoelastic behavior. The measurement of each sample storage modulus (G′) and loss modulus (G″) was performed. The sample strain was maintained at 0.1 N and the frequency varied from 0.01 to 10 Hz. The storage elastic modulus (G′), loss elastic modulus (G″), and loss tangent (tan δ) are, respectively:$$G^{\prime }=\frac{\sigma 0}{\varepsilon 0}COS\;\delta ,\;G\prime \prime =\frac{\sigma 0}{\varepsilon 0}\sin\delta ,\;\tan\delta =\frac{G\prime \prime }{G^{\prime }}$$

### 2.6. In Vitro of Magnesium Ion Release Experiment

The prepared 2.5 GMPC group, 5 GMPC group, and 7.5 GMPC group were placed in a 6-well plate and soaked with 5 mL PBS. Magnesium ion concentration was detected and recorded at the days 1, 3, 5, 7, 14, 21, and 28. We took 2 mL of sample liquid out, diluted the sample to 10 mL with PBS, and then analyzed it via inductively coupled plasma optical emission spectrometry (ICP, PerkinElmer, Waltham, MA, USA, Optima 7000DV). The concentration of Mg^2+^ in the solution was determined by referring to standard Mg^2+^ solutions. Three parallel measurements were conducted for averaging. To continue the release, fresh PBS (2 mL) was provided and the system was kept at 37 ± 1 °C for a longer period.

### 2.7. In Vitro of Cell Culture and Cell Viability Analysis

Human umbilical vein endothelial cells (HUVECs) and rat bone marrow mesenchymal stem cells (MSCs) were cultured in an incubator containing 5% CO_2_ at 37 °C. HUVECs were inoculated on GelMA, 2.5 GMPC, 5 GMPC, and 7.5 GMPC, with a density of 1 × 10^4^ cells/sample, and cultured for 3 days for live and dead staining, phalloidine staining, and DAPI staining. MSCs were inoculated on GelMA hydrogel with a density of 2 × 10^4^ cells/sample and cultured on 2.5 GMPC, 5 GMPC, and 7.5 GMPC. MSCs were cultured with 24 h sample extract, transwell experiments were performed 8 h later, and tube formation assay were performed 1 day later. Alkaline phosphatase (ALP) and ARS staining were performed at 15 and 21 days of culture. After 1 day of cultivation of MSCs using a special medium, the medium was changed to a DMEM-F12 induction medium. Then, we changed the medium every 2 days. Alkaline phosphatase (ALP) staining and ARS staining were performed at 15 and 21 days of culture, and the medium was changed every 2 days in all tests. MSCs were stained with ALP and ARS to observe the ALP activity and bone-induced differentiation of different groups.

### 2.8. In Vivo Study

There were 15 8-week-old male Sprague–Dawley (SD) rats with an average body weight of 180 g. The feeding conditions were 23 ± 2 °C, 50 ± 5% humidity, and 12 h light–dark cycle. All experimental procedures were carried out in accordance with the protocol approved by the Konkuk University Institutional Animal Care and Use Committee. During the procedure, general anesthesia was administered through an intraperitoneal injection of 8% chloral hydrate solution. The surgical area was shaved and sterilized with iodophor. For each rat, an instrument was used to create a critical skull defect of 6 mm diameter, symmetrical to the middle seam. The rats were randomly divided into 5 groups: negative control (defect only) group, GelMA hydrogel group, 2.5 GMPC group, 5 GMPC group, and 7.5 GMPC group. The GelMA hydrogel and GMPC sample were prepared into a disc shape with a diameter of 6 mm and a thickness of 1 mm. The periosteum and skin were sutured after implantation, and the animals were allowed to move freely in the cage with food and water. Animals were sacrificed 8 weeks after surgery.

### 2.9. Microcomputed Tomography (Micro-CT) Analysis

At 8 weeks after implantation, the skull was collected for micro-CT examination. The removed skulls were fixed in a 10% formalin solution for 24 h and subsequently stored in 70% ethanol at 4 °C until imaging with Scanco μCT 35 (Scanco Medical AG, Bruttisellen, Switzerland). The skull was soaked with PBS before scanning and placed directly onto the scanner’s sample holder. Imaging was performed using a medium-resolution setting with a source voltage E of 70 kVp, current I of 114 μA, and voxel size of 12.5 μm. We plotted the contours of the skull region to establish the volume of interest for quantifying mineralization, and used the optimal arbitrary thresholds of 20 (showing scaffold and mineralization) and 80 (mineralization only) to distinguish between truly mineralized and unmineralized skulls. Next, we performed 3D reconstruction analysis using the Scanco evaluation script. We exported the scan results for subsequent analysis using ImageJ (version 64-bit Java 8) For analysis, heterogeneity between micro-CT results in vivo was explained by describing the specific volumes used for comparison and by using correction factors for natural bone. For each slice scanned, a circular selection with a diameter of 6 mm (defect) was made, and a macro was used to measure the average gray value of each area. All slices with a negative average gray value were excluded. For each remaining slice, the median of the average gray value was calculated, and a total of 40 slices were analyzed with the median as the center. For the primary bone, an average of 40 pieces were used as internal control and correction factors for defects. The average grayscale value within each of the 40 slices of the defect was then divided by the correction factor for each animal to obtain the final value of the defect/primary mineralization. For quantitative analysis, new bone volume (mm^3^) and percentage bone volume ratio (bone volume/tissue volume, %) were measured.

### 2.10. Histological Analysis

The skull specimens were fixed with 10% neutral paraformaldehyde tissue fixative and decalcified with 10% EDTA solution for 1 month. After conventional treatment, the sample was embedded in paraffin wax and cut into 4 μm slices. Sections were examined via HE and Masson staining microscopy.

### 2.11. Statistical Analysis

Statistical analysis was performed using two-side difference analysis (ANOVA) and Tukey’s multiple comparison post hoc test. All values were expressed as mean ± standard deviation and compared between experimental groups.

## 3. Results

### 3.1. Characterization of GelMA Hydrogels and GMPC

#### 3.1.1. XRD and FTIR of GelMA and GMPC

The XRD patterns of MPC and GMPCs are shown in Figure 2A. Compared with MPC’s standard PDF card, the positions of these diffraction peaks in different samples were similar. A large number of MPC peaks were observed over the measurement range, indicating that the main hydraulic product was essentially similar, as were peaks of potassium magnesium phosphate hexahydrate (KMgPO_4_·6H_2_O).

The FTIR patterns of MPC and GMPCs are shown in Figure 2B. We found nearly identical functional groups in GelMA and GMPC samples, such as N-H (3282 cm^−1^), C-H (2927 cm^−1^), and C=O (1635 cm^−1^). The above can indicate that GelMA is present in all samples.

#### 3.1.2. Mechanical Properties and SEM of GelMA and GMPC

The sample was prepared by mixing GelMA hydrogel with MPC in three ratios. MPC was significantly displayed at 7.5 GMPC, while cross-linked GelMA hydrogels were observed to be translucent colloids between MPC powders as the proportion of mixed MPC decreased. The viscoelastic modulus of 5% GelMA hydrogel and GMPC samples were measured at 0.1–10 Hz. The storage moduli of 7.5 GMPC, 5 GMPC and 2.5 GMPC were 2.1483 ± 0.0164, 1.3880 ± 0.0318 and 0.8537 ± 0.0878 kPa, respectively (Figure 3B). In addition, the loss modulus was similar (0.067 ± 0.011 kPa) for all samples except 7.5 GMPC (0.0321 ± 0.0012 kPa). The loss tangent values of 2.5 GMPC and GelMA hydrogels were similar (Figure 3C).

The morphology of the samples was confirmed via FE-SEM (Figure 3A). MPC exists in the form of 200 µm particles. GelMA hydrogels exist in the form of sponges with pore diameters of 2–80 µm. As shown in the 2.5 GMPC, 5 GMPC, and 7.5 GMPC images, MPC was coated onto GelMA hydrogel. The spongy structure of GelMA hydrogel confirmed that the MPC’s location was fixed. A large amount of MPC was found in 7.5 GMPC.

### 3.2. Cell Viability of HUVECs on the GelMA Hydrogels and GMPCs

In order to compare and evaluate the HUVEC survival rate of 2.5 GMPC, 5 GMPC, 7.5 GMPC and CelMA, materials from the four groups were co-cultured with HUVECs for 3 days, respectively, and staining was performed with live or dead staining (Figure 4A), phalloidine staining, and DAPI staining (Figure 4B) to observe the differences among the groups. It can be seen that the cell activity of the 5 GMPC group was higher than that of other groups.

### 3.3. Osteogenic Differentiation of MSCs on the GelMA and GMPC

Transwell results of MSCs co-cultured with a composite extract are shown in Figure 5A. Experimental results of tube formation assay after 8 h co-culture of MSCs and material extract are shown in Figure 5B. The number of segments, nodes and total segments length are shown in the Figure 5E–G. Representative ARS staining of MSCs co-cultured for 21 days is shown in Figure 5D. Representative ALP staining of MSCs co-cultured for 15 days is shown in Figure 5C. In the above experiments, the result of 5 GMPC was significantly higher than that of other groups, and the results of GelMA and control group were similar. GMPCs can promote bone differentiation through gene expression of COL1, OPN, OCN and BSP (Figure 5H–K). These are genes involved in bone formation. In the results of 5 GMPC for 5 days, the expression of COL1 and OPN was 2.88 and 2.06 times that of the GelMA group, respectively. In the 5 GMPC group, OCN and BSP expression were considerably higher than other expressions at 15 days. On the other hand, gene expression in 7.5 GMPC and 2.5 GMPC tended to be similar to or slightly higher than that in the GelMA group.

### 3.4. Bone Regeneration of GelMA and GMPCs in an Animal Model

In vivo, GelMA hydrogel and GMPCs were implanted in the rat skull defect model. Reconstruction of the defect at 8 weeks after implantation is shown in the figure. Bone regeneration was more significant in the GelMA and GMPC groups than in the blank group, with 5 GMPC being the most significant. After 8 weeks, bone formation increased in each group, but not completely. In contrast, most of the defect area in the 5 GMPC group was filled with regenerated bone within 8 weeks (Figure 6A). Quantitative analysis of the bone volume fraction in the total tissue volume (BV/TV, %) in each group is shown in Figure 6B. As can be seen from the figure, the group of 5 GMPC (41.38%) was significantly higher than the other groups, and the GelMA group (21.27%) was also better than the control group (9.67%), indicating that GelMA alone may also help bone growth.

For histological examination, the defect was cut vertically at the center line of the defect area. In the defect group only, the defect area was full of fibrous connective tissue and only a small amount of bone was formed near the edge. In GelMA group, the new bone was well fused from the original edge of the bone defect, and small bone islands were visible. The residual GelMA hydrogel material was calcified and surrounded by newly formed ossified bone. In the GMPC groups, a large number of new bones were found to cover the old bone and grow towards the defect site, and the volume of regenerated bone in the 5 GMPC group was significantly higher than that in the other groups. Residual GelMA and MPC were mixed with the bone-like matrix and surrounded by newly formed bone.

As in the early regenerative stage of intramembrane ossification, in the GelMA and GMPC groups, osteoblasts line the outer surface of the newly formed bone surrounding GelMA and MPC. HE and Masson staining showed osteoid matrix and lamellar bone formation near the implant. HE staining and Masson staining were used to determine the new bone formation in the bone-filled area of the defect site (Figure 7). The figure shows obvious inflammatory cells and fibrous tissue, indicating long-term inflammation in the body after GMPC implantation. These inflammatory responses to some extent prevent the formation of new bone [33,34]. GelMA preserves protease degradation sites for collagen-containing RGD sequences of arginine, glycine, and aspartate. Therefore, the degradation rate of GelMA is relatively fast, generally occurring within 8 weeks. However, the animals in this study were sampled at week 8, and in the middle-to-late stages of in vivo experiments GMPCs may not be able to participate in promoting new bone formation due to complete degradation. The GelMA and GMPC groups showed a significant increase in the new bone area, and the increase was much larger in the 5 GMPC group.

## 4. Discussion

As shown in Figure 2A,B, XRD and FTIR results of GMPCs show that MPC peaks and GelMA radicals exist simultaneously [32,35]. This shows that, even after the composite preparation process, there is no change in the physicochemical property of these materials. Viscoelastic measurements in Figure 3B,C show that GMPC has stronger mechanical properties than GelMA. A large storage elastic modulus means that the material is close to being completely elastic, while material with a large loss elastic modulus can be in a viscous or water-like state with no elasticity. An increase in the loss tangent also means a decrease in elasticity [36,37]. In the SEM experiment in Figure 3A, MPC particles can be observed in the spongy structure of GMPCs. This indicates that the compressive strength of GMPCs increases with the increase in MPC content.

In vitro and in vivo experiments are shown in Figure 4, Figure 5, Figure 6 and Figure 7. The results reveal that 5 GMPC group was superior to other groups in cell activity, proliferation, and differentiation. Although GelMA group also showed some biocompatibility and osteogenic ability, there was still a significant gap with 5 GMPC. This shows that, in addition to GelMA, MPC plays a more critical role in GMPCs. After crosslinking GelMA and MPC, the degradation time was significantly shortened. At 8 w, the area of new bone in the experimental group increased significantly, the degradation of materials provided space for the growth of new bone, and the MPC particles remaining in situ could be further degraded to promote bone formation. It is worth noting that GelMA hydrogel itself has a good ability to promote bone regeneration and integrates well with the original bone. When GelMA was applied with MPC, these features were more pronounced, with significantly enhanced bone regeneration and increased expression levels of osteoblast markers.

The ideal material should maintain its shape and mechanical properties during bone regeneration. Moreover, the balance between the degradation rate of the material and bone formation rate is crucial for continuous new bone ingrowth [38,39]. The degradation rate and ion release curve of bioceramics are mainly determined using the chemical composition and physicochemical properties. Generally, lower crystallinity, high specific surface area, and smaller particle size lead to higher solubility and ion release rate [40,41,42]. There are two main degradation mechanisms of GMPC composites: (1) the rapid degradation process of hydrogel crosslinking chemical bond breaking and (2) slow hydrolysis and dissolution process of MPC hydration products [36]. The rapid degradation of the composite may be closely related to the rapid hydrolysis and high swelling rate of the hydrogel coated with the material. With the increase in MPC content, the porosity and pore size of GMPC composites decreased, and the degradation rate decreased significantly. The addition of metal ions to GelMA enhances the electrical activity of GMPCs [43], which may be part of the reason for the accelerated degradation of MPC and the release of Mg^2+^. Traditional employment of MPCs is mainly subject to the form of tiny plates, not being conducted fully enough to provide surface area for the surrounding bone tissues. In this study, we used MPC powders as an additive to improve the mechanical strength of GelMA, while efficiently delivering MPC powders into the bone defects. The 7.5 GMPC is better in compressive strength and microstructure, but 5 GMPC has better biocompatibility and osteogenic ability, which is more in line with our clinical needs. The above results may be related to the Mg^2+^ concentration released by MPC.

With the degradation of GMPCs, Mg^2+^ was released from the system and promoted bone formation after reaching a certain concentration [37,44]. In normal adult blood, the reference range of serum Mg^2+^ concentration is 0.65–1.05 mM, and according to clinical data, the critical dose tolerated by the human body in plasma is generally accepted to be 3.5 mM [45]. The risk of severe hypermagnesemia syndrome was significantly increased when the concentration of Mg^2+^ increased to 7.5 mM [46]. Current measurements are mostly performed in vitro to determine the appropriate concentration or therapeutic window of Mg^2+^ [27,47,48]. Using pure Mg implant extracts, 10 mM Mg^2+^ was found to be the critical concentration that did not affect cell proliferation, and 15 mM Mg^2+^ could be considered the overall safe dose (cell survival ≥ 75%) [49]. In the magnesium ion release experiment, the 2.5 GMPC group, 5 GMPC group, and 7.5 GMPC group reached dynamic equilibrium at 9.92 mmol/L, 12.86 mmol/L, and 18.47 mmol/L, respectively (Figure 6C). In vitro and in vivo experiments showed that the results of the 5 GMPC group were better than those of other groups, indicating that the magnesium ion concentration of 5 GMPC group may be more suitable for cell and tissue growth.

Studies have shown that magnesium ions promote osteogenesis in a concentration-dependent manner. Mg^2+^ directly affects the Notch activation of mesenchymal stem cells, but not the osteoblasts into which mesenchymal stem cells differentiate, suggesting that Mg^2+^ has a specific role in maintaining the dryness of mesenchymal stem cells rather than osteogenesis. In addition, Mg^2+^ at moderate concentrations significantly promotes bone cell maturation and enhances cell attachment to acellular bone surfaces. Previous studies have speculated that Mg^2+^ regulates a variety of signaling pathways at different stages of the differentiation of mesenchymal stem cells into osteoblasts. For example, Mg^2+^ is thought to promote the proliferation and osteogenic differentiation of MSC by sequentially activating MAPK/ERK and Wnt/β-catenin signaling pathways. In addition, the activation of transforming growth factor-β (TGF-β) and bone morphogenetic protein (BMP) signaling, which regulate mesenchymal stem cell differentiation during bone development, is associated with Mg^2+^ stimulation. In vitro data in the study confirmed the direct osteogenic effect of Mg^2+^ on MSC, especially at the beginning of osteogenic differentiation, but we noted that excess Mg^2+^ did not contribute to late osteoblast activity, especially the mineralization of the extracellular matrix [49]. In fact, previous studies have shown that Mg^2+^ additions need to be adjusted to the optimal range (<15 mmol/L) to achieve an osteogenic effect. Otherwise, the presence of high levels of Mg^2+^ may not be conducive to the osteogenic differentiation of osteoblasts. Our experimental results show that the concentration of magnesium ion in the 5 GMPC group is close to the appropriate concentration and that the performance is the most representative in all aspects, while the effect of the 7.5 GMPC group is not good. These conclusions further support previous studies, indicating that when the magnesium ion concentration is lower than 15 mmol/L, the osteogenic ability is enhanced with the increase in magnesium ion concentration, while when the magnesium ion concentration is higher than the appropriate concentration, the increase in magnesium ion concentration may weaken the osteogenic ability [50]. Compared with existing Mg-based materials such as Mg-ion screws and ceramic alloy materials, GMPCs can be cured after filling the bone defect in a fluid state and has good biocompatibility. For small-site defects of non-load-bearing bones, it may be possible to repair them by post-curing by injecting GMPCs (avoiding surgery), which is not only easy to perform, but also can reduce clinical costs. Large local releases of Mg^2+^ during bone remodeling may affect the crystallization and properties of hydroxyapatite bone minerals as Mg^2+^ is a potent inhibitor of hydroxyapatite crystal growth. Compared with Mg-based materials alone, GMPCs release lower Mg^2+^ content and inhibits bone mineral formation less.

Our study still has many limitations. First of all, the preparation of the composite material is based on GelMA and MPC as an additive, which is lacking in compressive strength and can only be used as a repair material for non-load-bearing bone defects. Secondly, the experiment lacks more precise grouping, and the experimental results are perhaps not the most appropriate concentration. Thirdly, it is currently difficult to control and monitor the release of magnesium ions in vivo. Fourthly, as a foreign substance, the implantation of bone cement inevitably triggers an immune response that manipulates the subsequent biological behavior of bone repair. The process of new bone formation induced by a successful bone implant consists of three successive stages [51]: an early stage of coagulation and acute inflammation, a transient stage of bone formation with chronic inflammation and osteogenesis, and a stage of bone remodeling. However, more inflammatory cells can be seen from our experimental results, indicating that the potential inflammatory response brought by the experimental material needs to be further improved. Last but not least, further research is needed on how the developed new materials can be translated into clinical applications.

## 5. Conclusions

We prepared GelMA/MPC composites for bone regeneration. The composite material could maintain the reconstructed shape of the bone defect without the need for additional equipment. The mechanical properties of the prepared material were stronger than those of gels used alone. This was confirmed by viscoelastic measurements. In vivo and in vitro, the biocompatibility and bone-induced differentiation of GMPCs could be proved. GMPCs had good bone regeneration ability in the rat model of skull bone defect. The biocompatibility of 5 GMPC was high and it had good bone regeneration ability. It could also maintain the proper shape and had good mechanical strength. Therefore, when applied as a bone filling material, the material was not only convenient for clinical use, but also could reduce the number of operations and improve patient satisfaction. In addition, there were many reviews on the therapeutic mechanism of Mg^2+^ and the research progress and application of biomaterials containing magnesium. Recent evidence suggests that the therapeutic effect of Mg^2+^ exhibits a significant concentration and stage-dependent behavior. Therefore, in future studies, we also need to explore the targeted administration and controlled release of Mg^2+^, as well as the association between GMPCs and inflammatory response, in order to truly meet the needs of clinical applications.

## Figures and Tables

**Figure 1 biomedicines-12-00228-f001:**
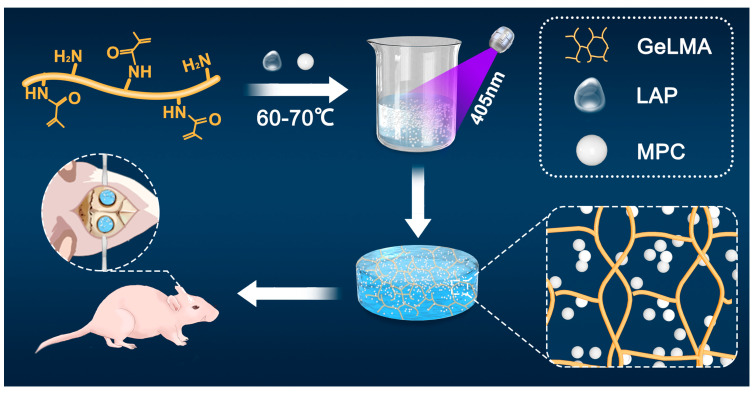
Production of GelMA/MPC (GMPC) composite and its application in the repair of bone defects.

**Figure 2 biomedicines-12-00228-f002:**
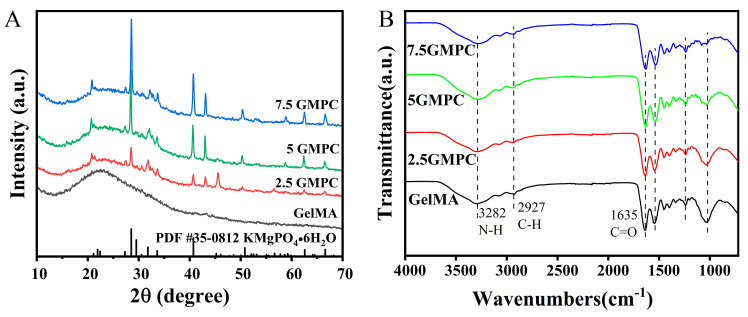
(**A**) The XRD spectra of GelMA: 2.5 GMPC, 5 GMPC and 7.5 GMPC. (**B**) FT-IR spectra of GelMA: 2.5 GMPC, 5 GMPC and 7.5 GMPC.

**Figure 3 biomedicines-12-00228-f003:**
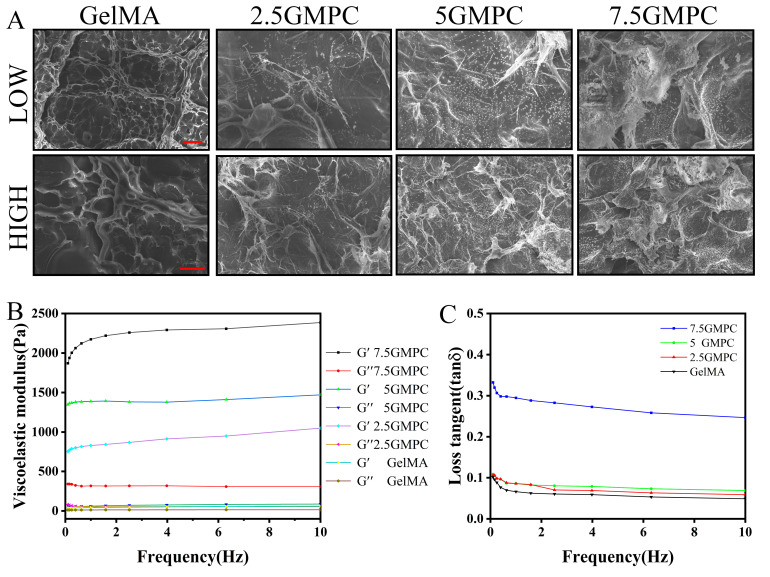
(**A**) SEM images of the sample were prepared. (**B**) Relation of storage modulus (G′) and loss modulus (G″) to GelMA: 2.5 GMPC, 5 GMPC and 7.5 GMPC frequencies. (**C**) The viscoelastic modulus of the sample is calculated and compiled to a loss tangent value. The red scale for low and high images is 500 μm and 50 μm respectively.

**Figure 4 biomedicines-12-00228-f004:**
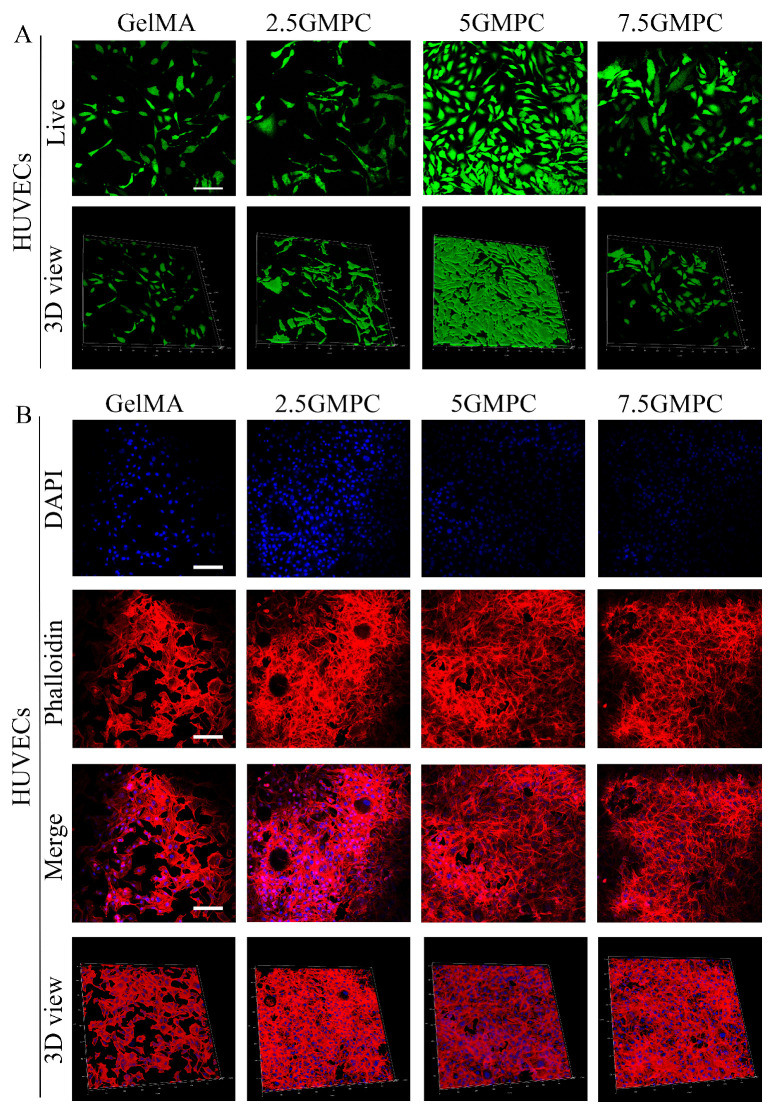
(**A**) The effect of GelMA and GMPC on HUVECs adhesion and proliferation. Staining was performed with live or dead staining: live cells were stained green and dead cells were stained red. (**B**) Phalloidine staining and DAPI staining: living and stationary cells are shown in blue.

**Figure 5 biomedicines-12-00228-f005:**
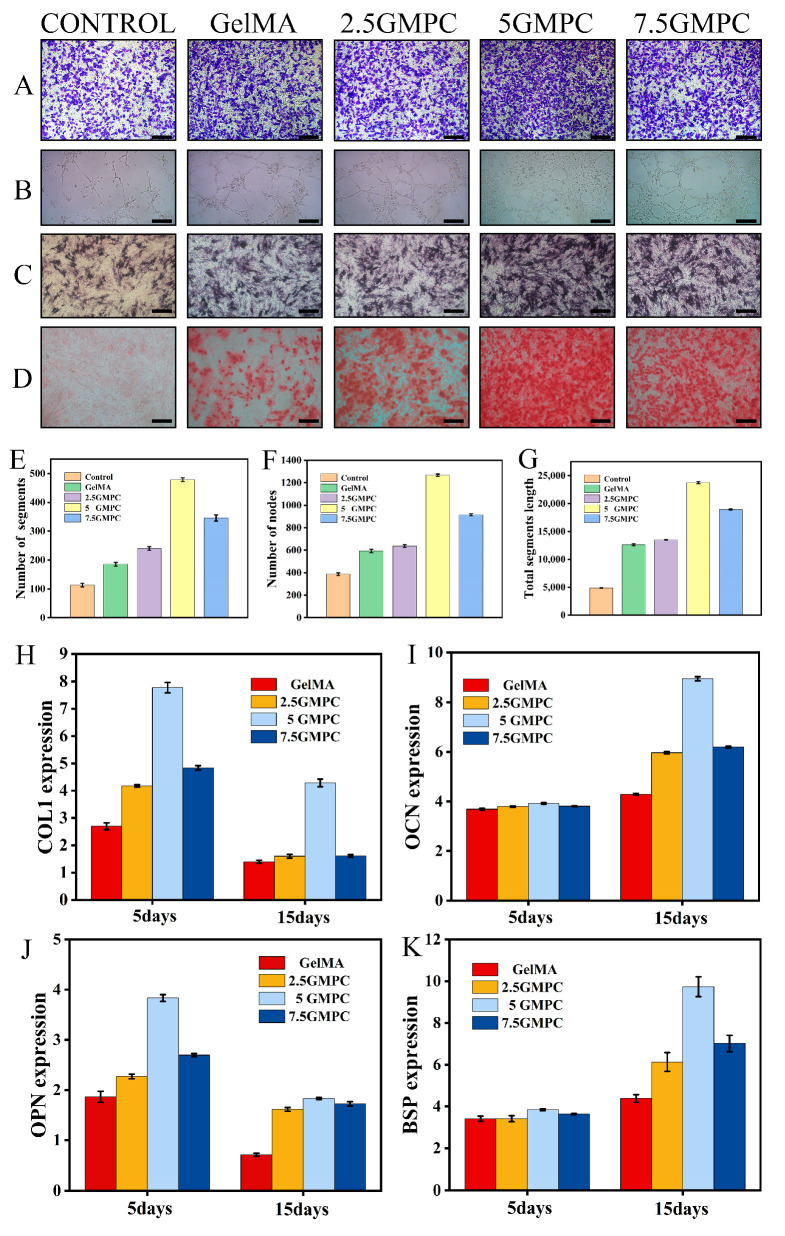
(**A**) Transwell results of bone marrow mesenchymal stem cells co-cultured with composite extract; (**B**) tube formation assay after 8 h co-culture of MSCs and material extract; and (**C**) representative ALP staining of bone marrow mesenchymal stem cells co-cultured for 15 days. (**D**) Representative ARS staining of bone marrow mesenchymal stem cells co-cultured for 21 days. (**E**–**G**) The number of segments and nodes, and total segments length in tube formation. (**H**–**K**) Expression of bone-related genes on the GelMA and GMPCs. Results are mean ± SD of triplicate experiments: *p* < 0.05, significant differences as compared with control (GelMA) group; *p* < 0.05, significant differences as compared with 5 GMPC group. All scales bars: 200 μm.

**Figure 6 biomedicines-12-00228-f006:**
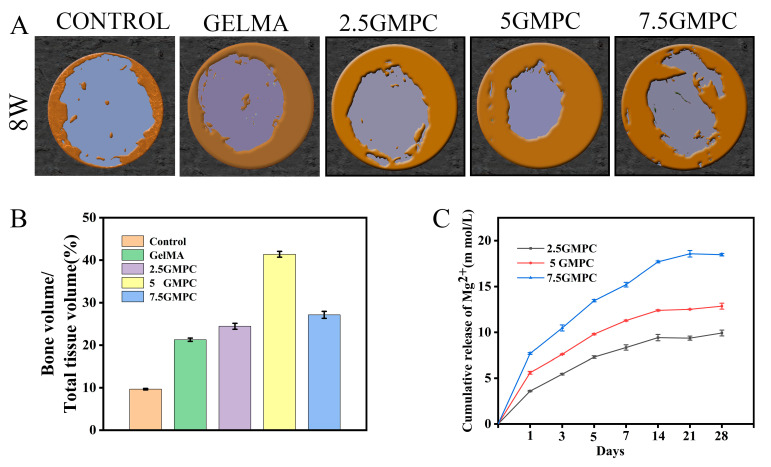
(**A**) Micro-CT analysis at 8 weeks post-implantation. Three-dimensional reconstructed micro-CT images of calvarial bone 8 weeks after implantation of control, GelMA and GMPC. (**B**) Quantitative analysis of bone volume fraction in total tissue volume (BV/TV, %) in each group. (**C**) Ion release of magnesium ions in the medium within 4 weeks.

**Figure 7 biomedicines-12-00228-f007:**
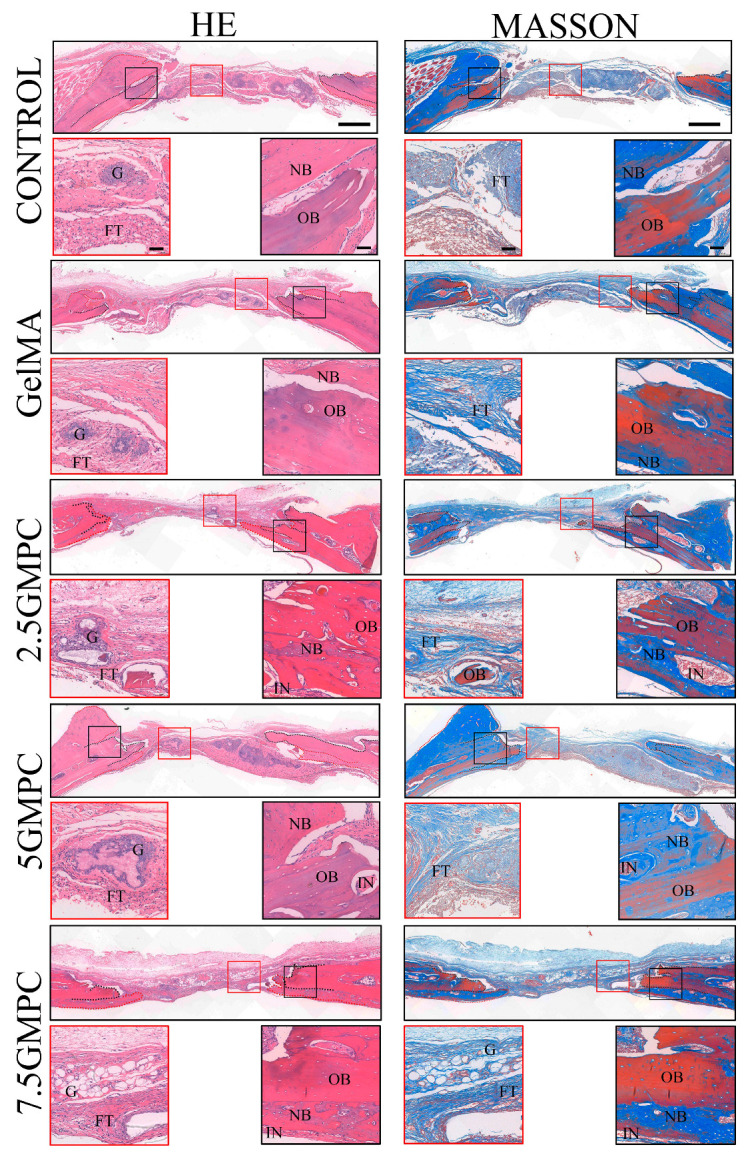
Histological analysis of defects at 8 weeks post-implantation. Representative images of histological sections in H&E stain, 8 weeks after implantation. Black-dotted line indicates the original edges of defects. Red-dotted line outlines regenerated bone. FT: fibrous tissue; G: GelMA; IN: inflammation; NB: newly formed bone; OB: original bone. The scale of the upper image is 500 µm and the scale of the enlarged image is 200 µm.

## Data Availability

Data are contained within the article.

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
