# Peer review of "A Bioactive Gelatin-Methacrylate Incorporating Magnesium Phosphate Cement for Bone Regeneration"

_biomedicines, 2024, doi:10.3390/biomedicines12010228_

Round 1

Reviewer 1 Report

Comments and Suggestions for Authors

The objective of this study was to address the deficiency of calcium phosphate cement by incorporating Gelatin-Methacrylate (GelMA) into magnesium phosphate potassium hexahydrate (KMgPO4‧6H2O), and the Magnesium Phosphate Cement Incorporating Gelatin-Methacrylate (GMPC) was prepared. Such recombination improves the mechanical strength of GelMA and modulates MPC's degradation and absorption cycles. GMPC exhibited a high survival rate and demonstrated a relatively high differentiation ability in bone regeneration. Furthermore, GMPC maintained its size and shape during the regeneration process, allowing for defect repair in various areas. Overall, the work of this paper is practical and logical. However, there are some problems to be further improved as well:

1. Bone cement usually has self-curing characteristics. The material in the thesis should be a porous dry gel of magnesium potassium phosphate and gelatin-methacrylate, which does not exhibit the self-curing cement characteristics of magnesium potassium phosphate or composite materials. Therefore, I think emphasizing magnesium phosphate cement is not a good choice. It is suggested that the title and keywords be optimized to reflect the paper's content better.

2. the methods and theoretical innovations are not significant in this paper. The author must dig deeper into the innovation points before accepting the article.

3. In the XRD pattern (Figure 2A), please add the standard line of the KMgPO4‧6H2O. Is the diffraction peak of 31.75 attributed to GelMA hydrogel? Please provide further evidence.

4. Please mark the size of the ruler clearly on the SEM images (Figure 3A).

5. In Lines 102-104, "MPC is mixed with 5% GelMA(w/v) by weight to volume ratio (g ml ml−1), resulting in 7.5%,5%, and 2.5% MPC solutions. Then, UV curing was performed by mercury arc lamp (405 nm) to prepare GelMA hydrogel." In Lines 209-210, "As shown in the 2.5GMPC, 5GMPC, and 7.5GMPC images, MPC is coated on GelMA hydrogel." Is there a problem with the methods or the microstructure description?

6. In Lines 217-218, "It can be seen that the cell activity of 5GMPC group was higher than that of other groups, and the blank group was close to that of GelMA group." Please give the results of the blank group in Figure 4.

7. In Lines 250-251, "In the GMPC group, the remaining MPC particles were clearly visible." Please mark the typical MPC particles in the figure.

8. Figure 6.(C) Ion release of magnesium ions in the medium within 4 weeks. Please add the test methods for this result.

9. The discussion section is divergent, and it is suggested that the main concerns of this work be focused on.

Comments on the Quality of English Language

The English of your manuscript must be improved before resubmission. I suggest that you obtain assistance from a colleague who is well-versed in English or whose native language is English.

Author Response

Thank you for your opinions and suggestions during your busy schedule. We have made relevant modifications according to your suggestions, please check.

1.Title has been changed to “A bioactive gelatin-methacrylate incorporating magnesium phosphate cement for bone regeneration ”. The key word has been changed to GelMA hydrogel /Bone regeneration/ MPC /Magnesium

2/9. The experimental method is supplemented and the discussion is optimized.

3.The XRD pattern has been improved (Figure 2A).

4.The red line in the SEM image is the scale (Figure 3A).

5.The writing order of 2.5GMPC group, 5GMPC group and 7.5GMPC group has been adjusted to be consistent. Lines 209-210 In the microscopic description of SEM, MPC particles can be seen attached to the surface of GelMA. There are no other errors in the experimental methods and microscopic descriptions.

6.In lines 217-218, the paper has been revised without further discussion because the blank group is of no practical significance.

7.In lines 250-251,it is an mistake in the writing process and has been deleted.

8.Figure 6.(C) Ion release of magnesium ions in the medium within 4 weeks. The experimental methods have been supplemented in 2.3.

Thank you again for your guidance. If we have other deficiencies, we hope to get more valuable comments from you.

Reviewer 2 Report

Comments and Suggestions for Authors

The submitted article showcases promising advancements in bone regeneration through GelMA/MPC composites, highlighting its mechanical strength and regenerative potential. However, several areas warrant consideration and further exploration for a more comprehensive understanding and future improvements.

• Limited Emphasis on Limitations: The article predominantly emphasizes the positive attributes of GelMA/MPC composites, but lacks detailed discussion on potential limitations or challenges faced during its application. A critical analysis of any drawbacks, such as long-term degradation or potential inflammatory responses, would provide a more balanced view for researchers and clinicians.

• In-Depth Understanding of Material Behavior: While the article reports enhanced mechanical properties of GMPC over GelMA alone, a deeper exploration into the underlying reasons behind these improvements could strengthen the study. A more comprehensive explanation of the material behavior, such as the influence of GelMA incorporation on MPC degradation kinetics or the interplay of gelatin and phosphate phases, would enhance the understanding of the composite's functionality.

• Detailed Insights from Animal Studies: The article briefly mentions the positive outcomes from animal experiments; however, a more detailed discussion on the limitations, if any, observed during these experiments could provide valuable insights. Additionally, incorporating data on long-term effects and potential adverse reactions in animal models could contribute to a more robust understanding of the composite's biocompatibility and efficacy.

• Potential for Controlled Release of Mg2+: The conclusion mentions the importance of exploring targeted administration and controlled release of Mg2+ in future studies. However, it would benefit from elaboration on the specific methodologies or strategies that could be employed for achieving controlled release. Further research in this direction could significantly enhance the clinical applicability and efficacy of the composite material.

• Addressing Clinical Application Challenges: The article highlights the convenience of GelMA/MPC as a bone filling material for clinical use. Nevertheless, discussing challenges related to scalability, cost-effectiveness, and compatibility with different bone defect types or sizes would provide a more comprehensive outlook for its potential clinical translation.

• References: The references list is quite short. Authors could benefit by incorporating the following references in the text along with other similar that they can find.

• 10.3390/nano13040692

• 10.3390/jfb14070341

• 10.3390/ijms242115748

• 10.1038/s41584-022-00887-0

• 10.3390/applmech2020018

• 10.1007/s13346-022-01222-6

In conclusion, while the study exhibits promising aspects of GelMA/MPC composites for bone regeneration, addressing these potential weaknesses and considerations could further advance the understanding, efficacy, and clinical applicability of this innovative biomaterial. Further research focusing on these areas may contribute significantly to its future development and utilization in the field of bone tissue engineering.

The reviewer.

Author Response

Thank you for your opinions and suggestions during your busy schedule. We have made relevant modifications according to your suggestions, please check.

1.A brief description of material limitations has been given at the end of the discussion.

2/3. A brief description of material limitations has been given at the end of the discussion and detailed Insights from Animal Studies :Your opinion in this regard is also considered by us before, but due to conditions, the two materials used in this study cannot react with each other. The future value of this composite material is not high, and it needs to be further improved in follow-up studies before it has the possibility of clinical application.

4.The control of magnesium ion slow release will be further studied in the future.

5.This study mainly explores the feasibility of the material, and solves the problems of this composite material one by one in the future research, so as to meet the expectations of clinical application.

6.The references have been refined based on your recommendations.

Thank you again for your guidance. If we have other deficiencies, we hope to get more valuable comments from you.

Reviewer 3 Report

Comments and Suggestions for Authors

In this manuscript, the author reported the preparation and characterization of bioactive glass/modified gelatin composite, and its application for bone regeneration, which is a meaningful work. Some issued need to be addressed before accepted.

(1) Extensive editing of format required. Eg. Line 10, Maintaining. Line 15 and 16, space should exist before cell and GMPC. Ling 74 and 76, GMPC neednt defined twice. Line 93-101, the MPCS, KH2PO4, ml-1 should revised.....

(2) Line 87-92, what is LAP? What is the difference of GelMA and GelMA-60? what is the molecular weight, grade, supplier of GelMA-60?

(3) For section 3.1.1, the XRD and FTIR of MPC should provided and discussed. Line 181, “The XRD patterns of MPC... and line 189, “The FTIR patterns of MPC”, the XRD and FTIR of MPC were not provided.

(4) Line 185, “GelMA hydrogel peaks were observed at 31.75°” is wrong, an obvious broad peak was observed around 22°, which should discussed with corresponding references.

(5) For the FTIR spectra, obvious peak at 3050 cm-1 for C=C (originated from methacrylate) can be observed, demonstrated the incompleted reaction of C=C, thus the reaction time should extended.

(6) Figure 3A should mentioned within manuscript.

(7) For figure 5E-K, the resolution should improved.

(8) Figure 6b and 6c, the font of sample name is inconsistent.

(9) References present in the conclusions should moved to the Discussions.

(10) As this investigation involving animals, the “Institutional Review Board Statement” should provided. In addition, the section of Funding, Data Availability Statement, Acknowledgments, and Conflicts of Interest should also added.

Comments on the Quality of English Language

Extensive editing of English language required

Author Response

Thank you for your opinions and suggestions during your busy schedule. We have made relevant modifications according to your suggestions, please check.

1.The article format has been improved.

2.LAP and GelMA-60 have been explained in Section 2.1.

3/4. The PDF card of MPC has been added to the XRD pattern. This XRD pattern only shows that MPC is contained in the composite material, so it is not explained too much.

5.GelMA was purchased by us from Engineering For Life, so the C=C spike in FTIR due to the synthesis process is not a factor we can control.

6.Figure 3A has been mentioned in the second paragraph of 3.1.2 "Mechanical properties and SEM of GelMA and GMPC".

7.The resolution of Figure 5E-K is consistent with that of other figures.

8.The fonts in Figure 6B and Figure 6C have been adjusted to match.

9.The references in the conclusions have been revised.

10.The relevant content has been added at the end of the article (lines 245-253).

Thank you again for your guidance. If we have other deficiencies, we hope to get more valuable comments from you.

Round 2

Reviewer 1 Report

Comments and Suggestions for Authors

As a second review, I do not think the discussion section is substantially improved. The authors have made most of the modifications according to the comments, and this manuscript could be considered for publication with minor revisions.

Comments on the Quality of English Language

The English language quality is generally good, but there is room for improvement. It would be beneficial to refine the language further, making it more precise and concise. This would enhance readability and make the language more impactful.

Author Response

Reviewer 1: 

As a second review, I do not think the discussion section is substantially improved.  The authors have made most of the modifications according to the comments, and this manuscript could be considered for publication with minor revisions.

Response: Thank you for your comments. We have revised the discussion part: (1) physicochemical properties and microstructure of GelMA and GMPCs; (2) Biocompatibility and osteogenic ability of GelMA and GMPCs; (3) The advantages of MPC Reviewer 1: powder as an additive; (4) Concentration dependence and mechanism of action of Mg2+; (5) Advantages and limitations of GMPCs (lines 296-394).

The specific changes are as follows:

  1. Discussion

As shown in Figure 2A-B, XRD and FTIR results of GMPCs show that MPC peak and GelMA radical exist simultaneously. This shows that even after the composite preparation process, there is no change in the physicochemical property of these materials. Viscoelastic measurements in Figure 3B-C show that GMPC has stronger mechanical properties than GelMA. A large storage elastic modulus means that the material is close to being completely elastic, while a large loss elastic modulus can be considered to be a viscous or water-like state with no elasticity. An increase in the loss tangent also means a decrease in elasticity. In the SEM experiment in Figure 3A, MPC particles can be observed in the spongy structure of GMPCs. This indicates that the compressive strength of GMPCs in-creases with the increase of MPC content.

In vitro and in vivo experiments showed in Figure 4-7 that 5GMPC group was superior to other groups in cell activity, proliferation and differentiation. Although GelMA group also showed some biocompatibility and osteogenic ability, there was still a significant gap with 5GMPC. This shows that in addition to GelMA, MPC plays a more critical role in GMPCs. After crosslinking GelMA and MPC, the degradation time was significantly shortened. At 8w, the area of new bone in the experimental group increased significantly, the degradation of materials provided space for the growth of new bone, and the MPC particles remaining in situ could be further degraded to promote bone formation. It is worth noting that GelMA hydrogel itself has a good ability to promote bone regeneration and integrates well with the original bone. When GelMA was applied with MPC, these features were more pronounced, with significantly enhanced bone regeneration and increased expression levels of osteoblast markers.

The ideal material should maintain its shape and mechanical properties during bone regeneration. Moreover, the balance between degradation rate of the material and bone formation rate is crucial for the continuous new bone ingrowth. The degradation rate and ion release curve of bioceramics are mainly determined by the chemical composition and physicochemical properties. Generally, lower crystallinity, high specific surface area, and smaller particle size lead to higher solubility and ion release rate. There are two main degradation mechanisms of GMPC composites :(1) The rapid degradation process of hydrogel crosslinking chemical bond breaking; (2) Slow hydrolysis and dissolution process of MPC hydration products. The rapid degradation of the composite may be closely related to the rapid hydrolysis and high swelling rate of the hydrogel coated with the material. With the increase of MPC content, the porosity and pore size of GMPC composites decreased, and the degradation rate decreased significantly. The addition of metal ions to GelMA enhances the electrical activity of GMPC, which may be part of the reason for the accelerated degradation of MPC and the release of Mg2+.Traditional employment of MPCs is mainly subject to the form of tiny plates, which is not fully conducted to enough surface area for the surrounding bone tissues. In this study, we used MPC powders as additive to improve the mechanical strength of GelMA, while efficiently delivering MPC powders into the bone defects. 7.5GMPC is better in compressive strength and microstructure, but 5GMPC has better biocompatibility and osteogenic ability, which is more in line with our clinical needs. The above results may be related to the Mg2+ concentration released by MPC.

With the degradation of GMPC, Mg2+ were released from the system and could pro-mote bone formation after reaching a certain concentration. In normal adult blood, the reference range of serum Mg2+ concentration is 0.65-1.05 mM, and according to clinical data, the critical dose tolerated by the human body in plasma is generally accepted to be 3.5 mM. The risk of severe hypermagnesemia syndrome was significantly increased when the concentration of Mg2+ increased to 7.5 mM. Current measurements are mostly per-formed in vitro to determine the appropriate concentration or therapeutic window of Mg2+. Using pure Mg implant extracts, 10 mM Mg2+ was found to be the critical concentration that did not affect cell proliferation, and 15 mM Mg2+ could be considered the overall safe dose (cell survival ≥75%). In the magnesium ion release experiment, 2.5GMPC group,5GMPC group and 7.5GMPC group reached dynamic equilibrium at 9.92mmol/L, 12.86mmol/L and 18.47mmol/L, respectively (Figure 6C). In vitro and in vivo experiments showed that the results of 5GMPC group were better than those of other groups, indicating that the magnesium ion concentration of 5GMPC group may be more suitable for cell and tissue growth.

Studies have shown that magnesium ion promotes osteogenesis in a concentration-dependent manner. Mg2+ directly affects Notch activation of mesenchymal stem cells, but not the osteoblasts into which mesenchymal stem cells differentiate, suggesting that Mg2+ has a specific role in maintaining the dryness of mesenchymal stem cells rather than osteogenesis. In addition, Mg2+ at moderate concentrations significantly promoted bone cell maturation and enhanced cell attachment to acellular bone surfaces. Previous studies have speculated that Mg2+ regulates a variety of signaling pathways at different stages of the differentiation of mesenchymal stem cells into osteoblasts. For example, Mg2+ is thought to promote proliferation and osteogenic differentiation of MSC by sequentially activating MAPK/ERK and Wnt /β-catenin signaling pathways. In addition, activation of transforming growth factor-β(TGF-β) and bone morphogenetic protein (BMP) signaling, which regulate mesenchymal stem cell differentiation during bone development, is associated with Mg2+ stimulation. In vitro data in the study confirmed the direct osteogenic effect of Mg2+ on MSC, especially at the beginning of osteogenic differentiation, but we noted that excess Mg2+ did not contribute to late osteoblast activity, especially the mineralization of the extracellular matrix. In fact, previous studies have shown that Mg2+ additions need to be adjusted to the optimal range (<15mmol/L) to achieve its osteogenic effect, otherwise, the presence of high levels of Mg2+ may not be conducive to osteogenic differentiation of osteoblasts. Our experimental results show that the concentration of magnesium ion in the 5GMPC group is close to the appropriate concentration and the performance is the most representative in all aspects, while the effect of the 7.5GMPC group is not good. These conclusions further support previous studies, indicating that when the magnesium ion concentration is lower than 15mmol/L, the osteogenic ability is enhanced with the increase of magnesium ion concentration, while when the magnesium ion concentration is higher than the appropriate concentration, the increase of magnesium ion concentration may weaken the osteogenic ability. Compared with existing Mg-based materials such as Mg-ion screws and ceramic alloy materials, GMPC can be cured after filling the bone defect in a fluid state and has good biocompatibility. For small site defects of non-load-bearing bones, it may be possible to repair them by post-curing by injecting GMPC (avoiding surgery), which is not only easy to operate, but also can reduce clinical costs. Large local releases of Mg2+ during bone re-modeling may affect the crystallization and properties of hydroxyapatite bone minerals, as Mg2+ is a potent inhibitor of hydroxyapatite crystal growth. Compared with magnesium-based materials alone, GMPC releases lower Mg2+ content and has less inhibition on bone mineral formation.

Our study still has many limitations. First of all, the preparation of the composite material is based on GelMA and MPC as an additive, which is lacking in compressive strength and can only be used as a repair material for non-load-bearing bone defects. Secondly, the experiment lacks more precise grouping, and the experimental results are maybe not really the most appropriate concentration. Thirdly, it is currently difficult to control and monitor the release of magnesium ions in the vivo. Fourthly, as a foreign substance, the implantation of bone cement inevitably triggers an immune response that manipulates the subsequent biological behavior of bone repair. The process of new bone formation induced by a successful bone implant consists of three successive stages: an early stage of coagulation and acute inflammation, a transient stage of bone formation with chronic inflammation and osteogenesis, and a stage of bone remodeling. However, more inflammatory cells can be seen from our experimental results, indicating that the potential inflammatory response brought by the experimental material needs to be further improved. Last but not least, further research is needed on how the developed new materials can be translated into clinical applications.

Reviewer 2 Report

Comments and Suggestions for Authors

The authors have not answered my comments extensively (only one sentence for some comments???) and they also have not incorporated all the references suggested. I instruct them to do so.

Comments on the Quality of English Language

Minor editing of English language required.

Author Response

Reviewer 2:

The submitted article showcases promising advancements in bone regeneration through GelMA/MPC composites, highlighting its mechanical strength and regenerative potential. However, several areas warrant consideration and further exploration for a more comprehensive understanding and future improvements.

Limited Emphasis on Limitations: The article predominantly emphasizes the positive attributes of GelMA/MPC composites, but lacks detailed discussion on potential limitations or challenges faced during its application. A critical analysis of any drawbacks, such as long-term degradation or potential inflammatory responses, would provide a more balanced view for researchers and clinicians.

Response: Thank you for your comments. The limitations and clinical challenges (Such as compressive strength, inflammatory response, magnesium ion release control and monitoring) of this material have been refined as required (lines 381-394 in green).

The specific changes are as follows:

Our study still has many limitations. First of all, the preparation of the composite material is based on GelMA and MPC as an additive, which is lacking in compressive strength and can only be used as a repair material for non-load-bearing bone defects. Secondly, the experiment lacks more precise grouping, and the experimental results are maybe not really the most appropriate concentration. Thirdly, it is currently difficult to control and monitor the release of magnesium ions in the vivo. Fourthly, as a foreign substance, the implantation of bone cement inevitably triggers an immune response that manipulates the subsequent biological behavior of bone repair. The process of new bone formation induced by a successful bone implant consists of three successive stages: an early stage of coagulation and acute inflammation, a transient stage of bone formation with chronic inflammation and osteogenesis, and a stage of bone remodeling. However, more inflammatory cells can be seen from our experimental results, indicating that the potential inflammatory response brought by the experimental material needs to be further improved. Last but not least, further research is needed on how the developed new materials can be translated into clinical applications.

In-Depth Understanding of Material Behavior: While the article reports enhanced mechanical properties of GMPC over GelMA alone, a deeper exploration into the underlying reasons behind these improvements could strengthen the study. A more comprehensive explanation of the material behavior, such as the influence of GelMA incorporation on MPC degradation kinetics or the interplay of gelatin and phosphate phases, would enhance the understanding of the composite's functionality.

Response: Thank you for your comments. There are two main degradation mechanisms of GMPC composites :(1) The rapid degradation process of hydrogel crosslinking chemical bond breaking; (2) Slow hydrolysis and dissolution process of MPC hydration products [doi: 10.1016/j.carbpol. 2022.119900]. The rapid degradation of the composite may be closely related to the rapid hydrolysis and high swelling rate of the hydrogel coated with the material. With the increase of MPC content, the porosity and pore size of GMPC composites decreased, and the degradation rate decreased significantly. The addition of metal ions to GelMA enhances the electrical activity of GMPC [doi: 10.1021/acsbiomaterials.0c01734], which may be part of the reason for the accelerated degradation of MPC and the release of magnesium ions (lines 322-329 in green).

The specific changes are as follows:

There are two main degradation mechanisms of GMPCs composites :(1) The rapid degradation process of hydrogel crosslinking chemical bond breaking; (2) Slow hydrolysis and dissolution process of MPC hydration products. The rapid degradation of the composite may be closely related to the rapid hydrolysis and high swelling rate of the hydrogel coated with the material. With the increase of MPC content, the porosity and pore size of GMPCs composites decreased, and the degradation rate decreased significantly. The addition of metal ions to GelMA enhances the electrical activity of GMPCs, which may be part of the reason for the accelerated degradation of MPC and the release of Mg2+.

Detailed Insights from Animal Studies: The article briefly mentions the positive outcomes from animal experiments; however, a more detailed discussion on the limitations, if any, observed during these experiments could provide valuable insights. Additionally, incorporating data on long-term effects and potential adverse reactions in animal models could contribute to a more robust understanding of the composite's biocompatibility and efficacy.

Response: Thank you for your comments. The long-term inflammatory response [doi: 10.1016/j.bone.2015.10.019] in vivo and the complete degradation of GMPC [doi: 10.1016/j.carbpol.2022.119900] reduced the rate of new bone formation, which did not achieve the desired effect (lines 281-288 in green).

The specific changes are as follows:

The figure shows obvious inflammatory cells and fibrous tissue, indicating long-term inflammation in the body after GMPC implantation. These inflammatory responses, to some extent, prevent the formation of new bone. GelMA preserves protease degradation sites for collagen containing RGD sequences of arginine, glycine and aspartate. Therefore, the degradation rate of GelMA is relatively fast, generally within 8 weeks. However, the animals in this study were sampled at week 8, and in the middle to late stages of in vivo experiments, GMPC may not be able to participate in promoting new bone formation due to complete degradation.

Potential for Controlled Release of Mg2+: The conclusion mentions the importance of exploring targeted administration and controlled release of Mg2+ in future studies. However, it would benefit from elaboration on the specific methodologies or strategies that could be employed for achieving controlled release. Further research in this direction could significantly enhance the clinical applicability and efficacy of the composite material.

Response: Thank you for your comments. In the vitro release experiment of Mg2+, it can be seen that at the same time point, the higher the proportion of MPC in GMPC, the higher the content of Mg2+ released, indicating that the release of Mg2+ in vitro experiment is controllable to a certain extent (lines 344-349 in green). However, it is difficult to control and monitor the release of Mg2+ (Example of ion-controlled release experimental method: core-shell bioceramic microspheres. It is possible to produce bilayered or multilayered microspheres with controllable shell-layer number and composition by varying the configuration of the capillary nozzles and the composition of the bioceramic slurries. [doi:10.1111/jace.14229]) (The internal environment is complex, there are many other ions interference), which is also a major problem faced by many researchers [doi:10.1016/j. jma.2021.03.004].

The specific changes are as follows:

In the Mg2+ release experiment, 2.5GMPC group,5GMPC group and 7.5GMPC group reached dynamic equilibrium at 9.92mmol/L, 12.86mmol/L and 18.47mmol/L, respectively (Figure 6C). In vitro and in vivo experiments showed that the results of 5GMPC group were better than those of other groups, indicating that the magnesium ion concentration of 5GMPC group may be more suitable for cell and tissue growth.

Addressing Clinical Application Challenges: The article highlights the convenience of GelMA/MPC as a bone filling material for clinical use. Nevertheless, discussing challenges related to scalability, cost-effectiveness, and compatibility with different bone defect types or sizes would provide a more comprehensive outlook for its potential clinical translation.

Response: Thank you for your comments. Compared with existing magnesium-based materials, GMPC can repair bone defects in a fluid state and has good biocompatibility. For small site defects of non-load-bearing bone, it can be repaired by curing after injection of GMPC (avoiding surgery), which is not only convenient to operate, but also can reduce clinical cost. Compared with magnesium-based materials, GMPC releases lower Mg2+ content and has less inhibitory effect on bone mineral formation [doi: 10.1016/j.actbio.2017.11.033], which can better meet clinical needs (lines 373-380 in green).

The specific changes are as follows:

Compared with existing Mg-based materials such as Mg-ion screws and ceramic alloy materials, GMPC can be cured after filling the bone defect in a fluid state and has good bio-compatibility. For small site defects of non-load-bearing bones, it may be possible to repair them by post-curing by injecting GMPC (avoiding surgery), which is not only easy to operate, but also can reduce clinical costs. Large local releases of Mg2+ during bone remodeling may affect the crystallization and properties of hydroxyapatite bone minerals, as Mg2+ is a potent inhibitor of hydroxyapatite crystal growth. Compared with Mg-based materials alone, GMPC releases lower Mg2+ content and has less inhibition on bone mineral formation.

References: The references list is quite short. Authors could benefit by incorporating the following references in the text along with other similar that they can find.

Response: Thank you for your comments. We have read and cited all the literature you have provided (2,3,7,8,25,50 in green).

The specific changes are as follows:

[2] G.N. Duda, S. Geissler, S. Checa, S. Tsitsilonis, A. Petersen, K. Schmidt-Bleek, The decisive early phase of bone regeneration, Nature Reviews Rheumatology 19(2) (2023) 78-95.

[3] D. Patel, S. Wairkar, Bone regeneration in osteoporosis: opportunities and challenges, Drug Delivery and Translational Research 13(2) (2023) 419-432

[7] A. Kantaros, T. Ganetsos, From Static to Dynamic: Smart Materials Pioneering Additive Manufacturing in Regenerative Medicine, International Journal of Molecular Sciences 24(21) (2023).

[8] A. Kantaros, D. Piromalis, Fabricating Lattice Structures via 3D Printing: The Case of Porous Bio-Engineered Scaffolds, Applied Mechanics 2(2) (2021) 289-302.

[25] J.Y. Wen, D.L. Cai, W.D. Gao, R.Y. He, Y.L. Li, Y.H. Zhou, T. Klein, L. Xiao, Y. Xiao, Osteoimmunomodulatory Nanoparticles for Bone Regeneration, Nanomaterials 13(4) (2023).

[50] M. Laubach, F. Hildebrand, S. Suresh, M. Wagels, P. Kobbe, F. Gilbert, U. Kneser, B.M. Holzapfel, D.W. Hutmacher, The Concept of Scaffold-Guided Bone Regeneration for the Treatment of Long Bone Defects: Current Clinical Application and Future Perspective, Journal of Functional Biomaterials 14(7) (2023).

In conclusion, while the study exhibits promising aspects of GelMA/MPC composites for bone regeneration, addressing these potential weaknesses and considerations could further advance the understanding, efficacy, and clinical applicability of this innovative biomaterial. Further research focusing on these areas may contribute significantly to its future development and utilization in the field of bone tissue engineering.

Response: Thank you for your valuable time and professional comments on this article.

Reviewer 3 Report

Comments and Suggestions for Authors

the revision is too simple, most of the issues were not resolved

Comments on the Quality of English Language

Extensive editing of English language required

Author Response

Reviewer 3:

In this manuscript, the author reported the preparation and characterization of bioactive glass/modified gelatin composite, and its application for bone regeneration, which is a meaningful work. Some issued need to be addressed before accepted.

Extensive editing of format required. Eg. Line 10, Maintaining. Line 15 and 16, space should exist before cell and GMPC. Ling 74 and 76, GMPC needn’t defined twice. Line 93-101, the MPCS, KH2PO4, ml-1 should revised.....

Response: Thank you for your comments. We have completed the modification of the article format (lines 9,16,85-93,134-142,164,172-173, in cyan).

Line 87-92, what is LAP? What is the difference of GelMA and GelMA-60? what is the molecular weight, grade, supplier of GelMA-60?

Response: Thank you for your comments. The introduction of LAP and GELMA-60 has been complete (lines 78-83 in cyan). Gelma-60 is a subbranch of GelMA, which is used in this study and then referred to as GelMA for short.

The specific changes are as follows:

GelMA was synthesized as previously reported. In brief, 20 ml of PBS were put in a brown bottle containing 0.05 g Lithium Phenyl (2,4,6-trimethylbenzoyl) phosphinate (LAP, Engineering For Life), followed by dissolving in a water bath at 40-50 °C for 15 minutes. Then, gelatin (GelMA-60 (Degree of amino substitution:60±5%, Molecular weight: 100-200kDa, Turbidity: <20%)) was dissolved in LAP solution with a concentration of 5% (w/v), followed by heating at 60-70℃ for 25 minutes in dark. Finally, basic GelMA solution (5%, w/v) was successfully prepared.

For section 3.1.1, the XRD and FTIR of MPC should be provided and discussed. Line 181, “The XRD patterns of MPC...” and line 189, “The FTIR patterns of MPC”, the XRD and FTIR of MPC were not provided.

Response: Thank you for your comments. A PDF card for MPC was added to the XRD pattern (Figure 2A). This XRD pattern is only used to show that the composite material contains MPC. Compared with GelMA, no new groups appear in the GMPC group in the FTIR diagram, indicating that the organic matter contained in GMPC is GelMA. Specific instructions have been added to the text (lines 204-211 in cyan). And we have added the equipment and methods for XRD determination (lines 104-110 in cyan).

The specific changes are as follows:

2.3. X-ray Diffraction

The XRD patterns were obtained using a X-ray diffractometer (Smart Lab SE, Japan). Film samples with dimensions of 5 mm ×5 mm were cut and fixed in a circular clamp of the instrument. The analysis was carried out directly and the conditions were as follows: (1) voltage and current: 40 kV and 40 mA, respectively; (2) scan range from 10°to 70°; (3) step: 0.1d°and (4) speed 1 d°/min, equipped with a secondary monochromator of graphite beam. The samples were stored at 25℃ and 50% Relative humidity (RH) and analyzed in triplicate.

Line 185, “GelMA hydrogel peaks were observed at 31.75°” is wrong, an obvious broad peak was observed around 22°, which should be discussed with corresponding references.

Response: Thank you for your comments. As for the GelMA peak at 31.75° in the XRD pattern, we found that this peak did not exist after re-detection. The possible cause is that the container of the material is not thoroughly cleaned, and trace impurities are mixed into it when GelMA is configured. Figure 2A and the error description in the text have been revised (lines 204-208 in cyan).

The specific changes are as follows:

The XRD patterns of MPC and GMPCs were shown in Figure 2A. Compare with MPC's standard PDF card, the positions of these diffraction peaks in different samples were similar. A large number of MPC peaks were observed over the measurement range, indicating that the main hydraulic product is essentially similar, and is potassium magnesium phosphate hexahydrate (KMgPO4·6H2O).

For the FTIR spectra, obvious peak at 3050 cm-1 for C=C (originated from methacrylate) can be observed, demonstrated the incompleted reaction of C=C, thus the reaction time should extended.

Response: Thank you for your comments. We have added the equipment and methods for FTIR determination (lines 111-116 in cyan). GelMA was purchased by us from Engineering For Life, so the C=C spike in FTIR due to the synthesis process is not a factor we can control. I hope you can understand this uncontrollable factor.

The specific changes are as follows:

2.4. Infrared Analysis

Chemical composition of GelMA and GMPCs were assessed by Fourier transform infrared spectroscopy with attenuated total reflectance (FTIR-ATR) using a FTIR 5700 (Thermo Electron Corporation, USA) spectrometer equipped with a diamond crystal at a nominal incidence angle of 45°and ZnSe lens. Spectra were recorded in the range of 600–4000 cm−1 at 32 scans with a resolution of 4 cm−1.

Figure 3A should mentioned within manuscript.

Response: Thank you for your comments. Figure 3A is referred to in lines 221-223(in cyan). And we have added the equipment and methods for SEM determination (lines 97-103 in cyan).

The specific changes are as follows:

lines 221-223:

The morphology of the samples was confirmed by FE-SEM (Figure 3A). MPC exists in the form of 200µm particles. GelMA hydrogels exist in the form of sponges with pore diameters of 2-80µm.

lines 97-103:

2.2. Microstructure Analysis.

For further experiments, the diameter of the crosslinked composite was cut to 6 mm. They are. The obtained GelMA, 2.5GMPC, 5GMPC and 7.5GMPC samples were freeze-dried in the freeze-drying machine for 36 h. The surface micro structure of GelMA and GMPCs were examined by high-resolution scanning electron microscopy (SEM) DSM 940 (Zeiss, Germany) after freeze-drying. Before examination, all specimens were stuck on special holders via conductive stickers and then sputtered with a thin (4 nm) gold layer for electron reflection.

For figure 5E-K, the resolution should improved.

Response: Thank you for your comments. The resolution of Figure 5E-K has been adjusted to be consistent with the other figures (the origin drawing is exported with a resolution of 600 dpi) (lines 230-231 in cyan).

The specific changes are as follows:

(E-G) The Number of segments, nodes and total segments length in tube formation. (H-K) Expression of bone related genes on the GelMA and GMPCs.

Figure 6b and 6c, the font of sample name is inconsistent.

Response: Thank you for your comments. The fonts in Figure 6B and Figure 6C have been adjusted to match (lines 255-257 in cyan).

The specific changes are as follows:

(B)Quantitative analysis of bone volume fraction in total tissue volume (BV/TV, %) in each group. (C) Ion release of magnesium ions in the medium within 4 weeks.

References present in the conclusions should moved to the Discussions.

Response: Thank you for your comments. We have moved the references in the conclusion to the relevant positions in the paper (references 27, 37,44,47,48) and added some references.

As this investigation involving animals, the “Institutional Review Board Statement” should provided. In addition, the section of Funding, Data Availability Statement, Acknowledgments, and Conflicts of Interest should also added.

Response: Thank you for your comments. We have added the content you requested in the article. (Author Contributions, Acknowledgments, Institutional Review Board Statement, Data Availability Statement (lines 411-419 in cyan).

The specific changes are as follows:

Author Contributions: Conceptualization, writing-review and editing, formal analysis, resources, X.Z.; investigation, project administration, methodology, C.G.; writing-original draft preparation, X.W.; visualization, Z.W.; supervision, funding acquisition, W.G. All authors have read and agreed to the published version of the manuscript.

Acknowledgments: The authors acknowledged the financial support from Science and Technology Department of Hubei Province, grant number 2020BCB058 from prof. Weichun Guo.

Institutional Review Board Statement: The animal study protocol was approved by the Institu-tional Review Board of Tongren Hospital, Wuhan University (Wuhan, China) (protocol codeSY2022-021).

Data Availability Statement: Data are contained within the article.

Round 3

Reviewer 2 Report

Comments and Suggestions for Authors

Thank you for your revisions.

Reviewer 3 Report

Comments and Suggestions for Authors

All of the issues mentioned were resolved in detail

Comments on the Quality of English Language

Moderate editing of English language required